# Season Long Pest Management Efficacy and Spray Characteristics of a Solid Set Canopy Delivery System in High Density Apples

**DOI:** 10.3390/insects10070193

**Published:** 2019-06-29

**Authors:** Paul Owen-Smith, John Wise, Matthew J. Grieshop

**Affiliations:** Department of Entomology, Michigan State University, East Lansing, MI 48824, USA

**Keywords:** spray coverage, spray deposition, pesticide application, SSCDS, tree fruit

## Abstract

Solid set canopy delivery systems (SSCDS) are a novel foliar agrochemical delivery system designed as an alternative for airblast sprayers in high density fruit production. This study tested the pest management potential, coverage, and chemical deposition of an SSCDS using commercially available microsprinkler components over the course of a growing season. Spray coverage and deposition for a representative airblast sprayer and SSCDS were evaluated using water sensitive paper and tartrazine dye, respectively. Foliar sprays for pest suppression were applied through both systems, and damage assessments were taken at the midpoint and end of the growing season. SSCDS sprays demonstrated similar levels of coverage on the adaxial leaf surface as airblast sprays, but significantly lower coverage on the abaxial surface. However, mean levels of foliar chemical deposition was generally higher in the SSCDS. Evaluations found minimal arthropod and fungal damage in both airblast and SSCDS treated plots compared to untreated trees. The SSCDS was shown to be a viable alternative to the airblast, with inherent advantages such as rapid application time and improved worker safety. Furthermore, higher deposition on SSCDS treated foliage supports the hypothesis that SSCDS provide a higher droplet capture rate in the canopy, with less off-target loss and drift than airblast sprayers.

## 1. Introduction

Solid Set Canopy Delivery Systems (SSCDS) represent a novel agrochemical application technology for high-density fruit production. SSCDS consist of a series of stationary microsprayers that are distributed throughout the orchard and fed from a common pumping station. This technology promises a rapid and precise method of chemical application, while removing personnel and heavy machinery from the orchard during pesticide applications. Immediate benefits of SSCDS include the minimization of worker pesticide exposure [1,2] as well as tree damage and soil compaction from heavy, tractor based sprayers [3,4,5]. Fixed emitter applications have been investigated sporadically for decades, beginning with the use of overhead frost protection sprinklers for agrochemical delivery, but these systems proved inefficient due to their reliance on a relatively small number of high volume emitters [6]. More recent research by Agnello and Landers [7] evaluated an SSCDS composed of a large number of low-cost plastic micro-sprinklers and pressure gated valves. This proof of concept has been expanded on by Sharda et al., and Owen-Smith et al., with results suggesting that a SSCDS could replace airblast spray application in high-density orchards [8,9,10].

The transition to high-density fruiting walls has resulted in a tall and narrow, planar tree architecture that is ill suited to applications by axial fan radial airblast sprayers [11,12,13]. This thin canopy profile, in conjunction with the high wind speeds associated with radial airblast sprayer design, commonly results in over-spraying of the lower and middle portion of the canopy and under-spraying of the top canopy portion [13,14,15]. Furthermore, large amounts of chemical applied by airblast sprayers are wasted when they are discharged into the local environment and atmosphere research as non-target drift [11,16,17,18,19]. These issues are exacerbated by sprayers that are poorly calibrated or optimized, with many growers applying a ‘one size fits all’ approach to the diverse canopy architectures presented by plantings of different training systems and phenological stage [13,20,21,22].

There are two main approaches to testing spray systems: observing pathogen and disease suppression and measuring spray deposits and coverage [23,24]. Deposition measurements determine the quantity of a chemical sprayed onto a surface, expressed in mass per area. Targets are sprayed with a tracer compound, and then collected and washed to recover the applied material. An effective method to quantify the recovered tracer utilizes absorption spectrophotometry and dye [25,26]. Coverage measurements describe the extent of the treated area, expressed as a proportion of the surface that receives treatment or contacts droplets [23]. Coverage demonstrates the extent of the treated area as well as the uniformity and quality of the spray, but lacks information on the total amount of chemical retained on target surfaces. Target cards that change color when contacted by droplets are placed within the canopy to mimic leaves, and then sprayed in a simulated application. Spray coverage has been used extensively in previous research into solid set canopy delivery systems [8,27,28] and airblast sprayers [20,23,29,30].

An SSCDS consisting of arrays of microsprinklers placed above and within a high-density apple canopy in Michigan USA provided the same level of pest management as an airblast sprayer. However, coverage evaluations performed as part of this experiment showed reduced coverage on the underside of leaves for the SSCDS compared with the airblast sprayer [10]. Studies conducted in high density apples in Washington State USA showed reduced coverage and deposition from an SSCDS compared to an airblast sprayer [31]. Similar experiments in Quebec, Canada and France evaluating SSCDS in high-density apples showed more heterogeneous spray deposits in SSCDS treated trees, yet both airblast and SSCDS had comparable efficacy in suppressing insect pests and diseases [28,32,33].

Based on the available literature it is clear that SSCDS provide more heterogeneous coverage especially on abaxial leaf surfaces compared to airblast sprayers but are capable of providing acceptable levels of insect pest and pathogen suppression. However, none of these studies have combined coverage, deposition and pest management efficacy measurements. Furthermore, the bulk of previous work either evaluated coverage at only one or two time points and in very small SSCDS systems. Thus, the specific objectives of this study were to:Quantify spray coverage on both the upper-side (adaxial) and under-side (abaxial) of leaf surfaces throughout a high density apple canopy at multiple time points.Quantify spray deposition on leaf surfaces at different levels within the canopy at multiple time points using a tartrazine tracer dye and absorption spectrophotometry.Evaluate season-long pest management of the SSCDS and its ability to suppress arthropod pests and plant pathogens compared to an airblast sprayer.

## 2. Materials and Methods

### 2.1. Experimental Area:

Experimental plots were established in a mixed variety apple (*Malus domestica*) orchard at Michigan State University’s Clarksville Research Center (CRC), in Clarksville, Michigan (42.8423° N, 85.2425° W). The five-year old planting consisted of 24, 137 m long rows, with eight rows each of three varieties ‘Crimson Royalty Gala’ on M.9 rootstock; ‘Honeycrisp’ on B.9 rootstock, and ‘Rubinstar Jonagold’ on B.9 rootstock. Trees were trained to the tall slender spindle system at a 0.9 m in-row spacing and 3.35 m between-row spacing. Planting density was 3720 trees ha^−1^.

Eight experimental units were established in the orchard: four airblast plots, and four solid set canopy delivery plots in a randomized complete block design. Each plot consisted of 306 trees – 102 of each variety. Plots were 0.09 hectares: nine tree rows wide and 34 trees long, established as squares that were approximately 30 m on each side. Each plot was surrounded by a 4.5 m buffer area on the north south axis to prevent spray movement down the row from one plot to another, and a 9 m buffer area on the east west axis to prevent spray movement across rows. No sampling points were on the periphery of the plots to ensure samples were unaffected by other treatments (Figure 1). All data were collected in 2016, coverage and deposition samples were collected on May 2nd, June 8th, July 12th, and August 8th. Weather conditions for coverage and deposition trials are summarized in Table 1.

### 2.2. Spray Systems

The equipment used in our experiment consisted of a SSCDS comprised of a canopy delivery system and a hydraulic/pneumatic pumping station and a radial fan airblast sprayer (Figure 2). The canopy delivery system was comprised of upper (2.5 cm) and lower (1.9 cm) Blue Stripe^®^ Poly Tubing polyethylene hoses (The Toro Company, Bloomington, MN, USA). Hoses formed a continuous loop, with the 2.5 cm line attached at 2.6 m high with the 1.9 cm line returning at 1.2 m high (Figure 2). The microsprinklers used in this study were Hadar 7110 series microsprinklers with ‘black’ 0.08 mm nozzles and ‘yellow star’ static spreaders (NaanDanJain Irrigation Ltd., Na’an, Israel). Individual components were fit together with bayonet style attachments, to prevent decoupling over the course of multiple duty cycles. An array of three microsprinklers were attached with 6.35 mm diameter tubing to a 240 kPa Leak Prevention Device (LPD) inserted into the upper delivery line (Figure 3). Each array of three microsprinklers consisted of a single microsprinkler attached to upper delivery line (2.6 m) and a pair of microsprinklers positioned at 1.2 m. The top microsprinkler was attached to the upper delivery line using a hose clip with a horizontal pillar that held the microsprinkler perpendicular to the row with microsprinklers spaced at 0.9 m intervals. The lower pairs of microsprinklers were mounted to a ‘T’ fitting and were spaced 0.9 m apart (Figure 3). The emitters on the top and bottom lines were offset such that they fell in between the two emitters in the opposing strata (Figure 2).

The SSCDS application equipment consisted of; a hydraulic pumping system, an air compressor, and a holding tank for spray material. Two tandem pumps powered by Honda GX160 160cc engines provided line pressurization and chemical delivery, run together to attain higher line pressures than a single pump could achieve. A diesel air compressor (D185PJD Sullivan Palatek, Michigan City, IN, USA) was used for system clearing and cleaning, and spray formulation was mixed and held in a 1892.7 L tank on the REARS Powerblast Sprayer (REARS MFG. CO., Coburg, OR, USA) used for the airblast comparison. The Rears airblast sprayer was outfitted with 23CER ceramic cores and DCER4 ceramic orifice discs, and was calibrated to apply a spray volume of 655 L ha^−1^.

Pest management, deposition and coverage applications were made using a single spray tank mixed in the airblast sprayer holding tank with airblast applications following SSCDS applications. The SSCDS was operated in a 5-step process. Step 1: spray material was mixed in the airblast sprayer holding tank. Step 2: spray material was pumped into the delivery system at <240 kPa until all lines were full as indicated by fluid returning to the tank through the return valve. Step 3: the return valve to the tank was shut and pump pressure increased to 415 kPa for a 10 s interval applying a spray volume of 655 L ha^−1^ (132.36 mL/microsprayer). Step 4: return valves to the tank opened and compressed air run into the system <240 kPa to return excess material to the tank. Step 5: return valves to tank shut and compressed air run into the system at 415 kPa to purge remaining material in nozzles. Three rows were sprayed at a time within a block to minimize pressure loss across the system. Spray volume was confirmed with a volumetric test that demonstrated average nozzle output was 132.4 mL.
Spray material mixed in holding tank.Liquid mix is drawn out of tank by tandem gas powered pumps.Mix is pumped through manifold at <240 kPa into 5 cm diameter PVC header linesOnce PVC lateral lines are filled, mix moves up into 2.5 cm polyethylene delivery lines.Mix fills top line, and then begins to return in 1.9 cm bottom hose.Circulated liquid is returned to holding tank and lines are filled with spray material.Return valve is closed, pump pressure increased to 415 kPa.Mix is applied through microsprinklers for 10 s (655 L ha^−1^).Return valve is opened, and air compressor pressurizes line at <240 kPa, pushing excess spray back to holding tank.Air pressure is increased to 415 kPa to purge mix from microsprinklers.

### 2.3. Coverage, Deposition, and Insect and Disease Damage Evaluations

#### 2.3.1. Coverage

Coverage was recorded and quantified using 26 × 52 mm water sensitive paper (WSP) cards (TeeJet^®^, Spraying Systems Co., Wheaton, IL, USA) which change color when wetted. Four trees per plot were sampled in May, with 8 trees per plot sampled in June, July, and August. Cards were placed on both sides of the tree respective to the row at three different vertical locations: ‘low’, 0.7 m above ground; ‘medium’, 1.4 m; and ‘high’, 2.1 m. Heights were as close as the branching of the canopy would allow, within approximately 0.15 m of the target height. Cards were clipped in pairs onto the adaxial and abaxial leaf surfaces (face up, face down) with binder clips. Cards were collected after drying in situ (30–60 min) and affixed to a sheet of paper pre-labeled with date, row side, row number, plot number, collection height, card orientation and treatment. Coverage evaluations were done on days with <80% relative humidity in the afternoon and similar ambient conditions. Once cards were collected they were stored in a sealed container with silicon desiccant to prevent any issues with water vapor reacting with the coating.

A flatbed scanner was used to digitize the cards and associated identification for further analysis at a 1200 dpi resolution. ImageJ software [34], using the plugin DepositScan to automatically calculate the percentage card coverage [35]. The software converts the image to greyscale and uses the intensity values of each pixel to determine the area of the droplets that have reacted with the surface coating of the card. However, DepositScan occasionally had difficulties recognizing droplet coverage on cards with <5% coverage and >95% coverage and returned inaccurate coverage values in those ranges. When the entire card is close to uniform in color, the automatic threshold for pixel values determined to be droplets is thrown off, and DepositScan will set an erroneous threshold for pixel selection. For these cards, a manual threshold slider was used to differentiate the droplets from the background.

#### 2.3.2. Deposition

Deposition was computed by determining the quantity of Keyacid Tartrazine (Keystone Corp., Chicago, IL, USA), a food grade yellow dye, recovered from 90 mm filter paper and leaf surfaces. Tartrazine was chosen due to its use in previous studies, low photodegradation rate, high recovery efficiency from targets, and lack of toxicity [25]. All coverage/deposition applications were made at a rate of 655 L ha^−1^ with a mix of 1 g L^−1^ Tartrazine and 1 mL L^−1^ of the nonionic surfactant Latron-B 1956. 30–60 min following spray applications, three mature leaves were collected from each sampling location once the application had dried. Leaf samples were collected from the same terminal where the WSPs were located, and stored in individual Ziploc bags. For residue anlaysis, a single leaf from each subsample location was placed in a 50 mL centrifuge tube (Denville Scientific Inc., Holliston, MD, USA), filled with 25 mL of water, and inverted repeatedly for 20 s. The leaf was then removed and placed in a leaf press with identifying information. Leaves were pressed and dried, and then a LI-COR 3100 Area Meter (LI-COR Biosciences, Lincoln, NE, USA) was used to determine the surface area. Two 200 µL samples were taken from each 25 mL sample washed from the leaves, and pipetted into the 96 well plate. Absorbance values were averaged and then used to calculate the concentration and therefore mass of tartrazine in each sample washed from the leaves. Total tartrazine mass was combined with the leaf surface area and deposition was expressed as µg/cm^2^.

Filter papers were used as artificial targets instead of leaves during the May sampling date due to insufficient quantities of leaves. A 96 well plate Biotek^®^ Synergy™ HT microplate reader (Fisher Scientific, Pittsburg, PA, USA) was used to ascertain the absorbance of samples at 435 nm. A standard curve was generated to determine the relationship between concentration and absorbance, as well as the minimum and maximum absorbance values. Serial dilutions were made from a stock solution of 1 g L^−1^ in concentrations of 1 g L^−1^, 0.5 g L^−1^, 0.1 g L^−1^, 0.075 g L^−1^, 0.05 g L^−1^, 0.025 g L^−1^, 0.01 g L^−1^, 0.0075 g L^−1^, 0.005 g L^−1^, 0.0025 g L^−1^, and 0.001 g L^−1^. The stock solution was used to make dilutions to 0.5, 0.1, 0.05, 0.01, 0.005 and 0.001 g, with each successive dilution made from the previous concentration, while the 0.075, 0.025, 0.0075 and 0.0025 g L^−1^ samples were made from the previous concentration but were not used to make the next dilution. Five 200 µL samples of each concentration were transferred to a 96 well plate with five water blanks, and an absorbance reading at 435 nm was taken. Data was then transferred to excel for organization and SAS for analysis with PROC REG. The magnitude of the absorbance for the 1 g L^−1^ and 0.5 g L^−1^ samples was too high for the microplate reader, but absorption values for concentrations between 0.1 g L^−1^ and 0.001 g L^−1^ showed a linear relationship with an R^2^ value of 0.996. Two replications with 5 subsamples of each concentration were used to create a linear regression, which was then used to produce an equation to calculate unknown concentrations from known absorbance values.

#### 2.3.3. Pest Management Efficacy

Pathogens and arthropod pests were managed throughout the spring and summer of 2016 with agrochemicals applied to the 0.09 ha plots through the solid set canopy delivery system and airblast sprayer (Figure 1). Spray formulations were applied on the advice of the Clarksville Research Station Assistant Farm Manager, and were identical to sprays applied to other orchards at the field station. Buffer rows and buffer gaps between plots were not treated. Each plot received the same treatment on the same day, with SSCDS plots sprayed first, and then airblast plots sprayed with the remaining tank mix. Plant protectants were applied at the same volume as sprays for coverage and deposition quantification. The first four fungicide sprays of the year (before 29 April, 2016) were only applied through the airblast, as the SSCDS was not yet de-winterized and running. Copper Hydroxide was applied at 6.75 kg ha^−1^ on 30 March, 2016, Mancozeb (Penncozeb, United Phosphorous, King of Prussia, PA, USA) at 3.35 kg ha^−1^ on 15 April, 2016, Captan (Captan 80 WDG, Arysta LifeScience, Cary, NC, USA) at 3.75 kg ha^−1^ on 20 April, 2016, and Manzoceb and Tebuconazole (Indar, Dow AgroSciences, Indianapolis, IN, USA) at 4.9 kg ha^−1^ and 0.35 L ha^−1^ on 25 April, 2016. Agrochemicals applied through both the system and the airblast for the rest of the season are summarized in Table 2.

Insect and pathogen damage were recorded in the mid-season and pre-harvest intervals, on 6 July, 2019 and 26 August, 2019 respectively. Assessments were made on the sixteen trees in each plot (two trees per sampling location indicated in Figure 1). To rate foliar apple scab (*Venturia inaequalis*, Cooke) severity, 20 terminals and 20 clusters were checked per tree, with half of the observations coming from each side of the row. Clusters and terminals were checked at all heights throughout the tree, and randomly selected. To estimate the abundance of fruit feeding pests and rate apple scab, 20 fruit on each tree from both sides of the row and throughout the height of the canopy were randomly selected and examined. Undergraduates assisting with damage assessment were all trained together and practiced until they repeatedly reached consensus on scoring apple scab lesions. Pest damage was at such low levels that any scars or marks that were unknown could be assessed by an expert.

Damage from apple scab was assessed as a percentage of the individual fruit’s surface area covered in visible lesions, approximated as either 0, 2, 5, 10, 20, 40, 60, 80, 90, 95, 98, or 100%. Data collected was then analyzed on the percent incidence of scab, which is the percent of fruit showing any amount of scab damage out of the total fruit checked in the plot. Damage from obliquebanded leafroller (*Choristoneura rosaceana*, Harris) was recorded from feeding in rolled leaves or leaf rolls or structures webbed upon fruit. Stinkbug (Pentatomidae) feeding injury assessed based on the distinctive conical pits and puncture marks the stylet produces. Codling moth (*Cydia pomonella*, L.) and Oriental fruit moth (*Grapholita molesta*, Busck) damage to fruit were recorded based on stings and larval entry tunnels and grouped as internally feeding pests. Plum curculio (*Conotrachelus nenuphar*, Herbst) counts were based on the pits left from feeding and the crescent shaped oviposition scar left on the skin of the apple following oviposition. Signs of insect damage were expressed as counts per structure- whether on a terminal, cluster, or fruit. Observers were trained using online extension materials provided at (https://www.canr.msu.edu/apples/pest_management/) and with reference samples collected on the day of evaluations.

Trees in four rows located in a separate high-density apple orchard 0.5 km away were assessed for the same insect and fungal damage. Trees in this orchard were at their 8th leaf and consisted of Gala planted on M9 with a 1 m interspace and 3.7 m row spacing. Trees in the check block were not treated with pesticides over the course of the season and served as a comparison to determine the relative potential for damage by pest arthropods and pathogens in the area. Previous pilot studies comparing the SSCDS and airblast demonstrated that untreated controls had orders of magnitude higher levels of pathogen and insect damage. Since this study was designed to directly compare airblast and SSCDS efficacy, the comparison with an untreated control was less important than maximizing the size and number of treated plots. An unreplicated check plot the same size as the experimental plot, and sampled in the same way, was used to confirm that ambient conditions resulted in commercially unacceptable levels of damage. The pest damage in the check plots was not used in any true statistical analysis, simply for comparison of means and standard error.

#### 2.3.4. Statistical Analysis

Adaxial and abaxial coverage data were arcsine transformed to meet assumptions of normality and to reduce heteroscedasticity, and analyzed separately. Both data sets still displayed significant heteroscedasticity for the factor ‘Treatment’ when tested with a Levene’s test so variances were grouped by treatment and analysis proceeded with an unequal variance model. The main fixed effects were date, treatment, and height, with plot as a random factor. Data were fit to a repeated measures split-plot ANOVA model in SAS 9.4 PROC MIXED (SAS Institute Inc., Cary, NC, USA) as each measurement at each date was taken from the same tree and height. Sample height was considered the subplot factor. Data from either side of the tree were not pooled, to prevent artificial variance reduction. There was a single missing data point in the adaxial coverage data set, and a pair in the abaxial coverage measurement. Residuals for each factor were and combination of factors were visually inspected with boxplots, residual vs. predicted value, and residual vs. quantile plots. Multiple comparisons for main effects and interactions were performed using the LSMEANS statement adjusted with Tukey’s HSD and Tukey-Kramer test Alpha levels were set to 0.05 throughout the analysis. Arcsine transformed least square means and standard errors were back transformed.

Coefficients of variation were calculated for each height of each plot for the three balanced dates as an index of dispersion to assess differences in variability displayed by the coverage profile of each spray type. Coefficients of variation were assessed for normality before being analyzed with a repeated measures ANOVA as well. Adaxial coefficients of variation showed no significant differences in variances with a Levene’s test, and met assumptions of normality, but abaxial coefficients of variation were square root transformed to meet normality assumptions.

Deposition was analyzed with an identical model to the abaxial and adaxial coverage data, but with variances grouped by date, treatment, and height in order to achieve the best model fit. For deposition data, 5 outliers were removed that were skewing means, with the cutoff at 5.00 ug/cm^2^. Four outliers came from SSCDS plots, and one was from airblast treated plots. Coefficients of variation were also computed for deposition and analyzed as before. Pest management data was non-normal and could not be transformed to meet normality or homoscedasticity assumptions. Counts of pest damage from each treated plot were compared between treatments with Wilcoxon rank sum tests using SAS 9.4 NPAR1WAY.

## 3. Results

### 3.1. Adaxial Coverage

A repeated measures ANOVA on data from the three months with balanced comparisons (June, July, and August) showed a significant difference in adaxial (upper surface) coverage attributed to the main effect of treatment (F_1,6_ = 15.92, *p* = 0.0072), and the interaction of treatment and height fixed effect terms (F_2,12_ = 5.66, *p* = 0.0186). Airblast treated plots had significantly higher coverage overall, and significantly higher coverage at the highest height on each date (Figure 4). Date was also a significant fixed effect (F_2,44_ = 9.13, *p* = 0.0005). Multiple comparisons by month using the Tukey-Kramer adjustment show that airblast treated plots displayed significantly higher coverage in one month, August (t_α = 0.05,44_ = 3.20, adj. *p* = 0.0289), while in June and July they were not significantly different at the alpha = 0.05 level (t_α = 0.05,44_ = 1.87, adj. *p* = 0.4359; t_α = 0.05,44_ = 2.89, adj. *p* = 0.0625) (Figure 4).

Comparisons between the two treatments at the same height showed a significant difference with greater airblast coverage on the highest sampled leaves (t_α = 0.05,12_ = 5.07, adj. *p* = 0.0029), but cards at the lower and middle heights did not show any significant differences (t_α = 0.05,12_ = 0.85, adj. *p* = 0.9517; t_α = 0.05,12_ = 2.03, adj. *p* = 0.3816). Different heights in airblast treated plots were not significantly different from each other: high and low (t_α = 0.05,12_ = 0.12, adj. *p* = 0.9046), high and middle (t_α = 0.05,12_ = −0.59, adj. *p* = 0.9896), and middle and low (t_α = 0.05,12_ = −0.71, adj. *p* = 0.9764). In comparison, different heights in SSCDS treated plots showed significant differences at the high and low heights (t_α = 0.05,12_ = −3.83, adj. *p* = 0.0226), but not at the high and middle (t_α = 0.05,12_ = −3.21, adj. *p* = 0.0641) or middle and low (t_α = 0.05,12_ = 0.62, adj. *p* = 0.9973).

Coefficients of variation for abaxial coverage were calculated for each plot for the June, July and August sampling dates. Application technology yielded a significant difference (F_2,6_ = 127.84, *p* < 0.0001), as well as height (F_2,12_ = 10.36, *p* = 0.0024), and date (F_2,36_ = 8.28, *p* = 0.0011). Coefficient of variation least square means were 0.5127 for the airblast and 0.9889 for the SSCDS, with a shared standard error of ± 0.029. Comparisons between least square means of coefficients of variation using Tukey’s adjustment were significantly different from each other in each of the three months, with a higher SSCDS σ²/µ at each date: June (t_α = 0.05,36_ = −3.38, adj. *p* = 0.0201), July (t_α = 0.05,36_ = −5.86, adj. *p* < 0.0001), and August (t_α = 0.05,12_ = −5.41, adj. *p* < 0.0001). Neither of the treatments displayed any significant differences with themselves across months-except for the SSCDS-where a significant difference occurred in σ/µ between July and August (t_α = 0.05,36_ = −3.51, adj. *p* < 0.0144) (Table 3).

### 3.2. Abaxial Coverage

A repeated measures ANOVA on data from the three months with balanced comparisons (June, July, and August) showed a significant difference in abaxial (lower surface) coverage by treatment (F_1,6_ = 200.72, *p* < 0.0001) with significantly higher coverage from the airblast. The interaction of treatment and height fixed effect terms (F_2,12_ = 12.88, *p* = 0.0010) was also significant. Date was also a significant fixed effect (F_2,44_ = 14.6, *p* < 0.0001). Multiple comparisons by month using the Tukey-Kramer adjustment show that airblast treated plots displayed significantly higher coverage in all three months: June (t_α = 0.05,44_ = 7.86, adj. *p* < 0.0001), July (t_α = 0.05,44_ = 10.18, adj. *p* < 0.0001), and August (t_α = 0.05,44_ = 10.48, adj. *p* < 0.0001) (Figure 5).

Comparisons between the two treatments at the same height yielded significant differences between at all three heights, with consistently higher coverage from the airblast sprayer: high (t_α = 0.05,12_ = 12.57, adj. *p* < 0.0001), middle (t_α = 0.05,12_ = 6.07, adj. *p* < 0.0001), and low (t_α = 0.05,12_ = 9.89, adj. *p* < 0.0001). Different heights in airblast treated plots were not significantly different from each other: high and low (t_α = 0.05,12_ = 3.1, adj. *p* = 0.0774), high and middle (t_α = 0.05,12_ = 1.82, adj. *p* = 0.4899), and middle and low (t_α = 0.05,12_ = 1.27, adj. *p* = 0.7942. In contrast, different heights in SSCDS treated plots yielded significant differences between the middle and low heights (t_α = 0.05,12_ = −4.99, adj. *p* = 0.0048) as well as between the middle and high heights (t_α = 0.05,12_ = −4.99, adj. *p* = 0.0033), but not between the high and low heights (t_α = 0.05,12_ = −0.23, adj. *p* = 0.9999) (Figure 5).

Coefficients of variation for abaxial coverage were calculated for each plot for the June, July and August sampling dates. The fixed effect of application technology was significant (F_2,6_ = 197.67, *p* < 0.0001), with higher coefficients of variation for the SSCDS treatments. Interaction of date*treatment and treatment*height were also significant (F_2,36_ = 5.45, *p* = 0.0085; F_2,12_ = 16.12, *p* = 0.0004). Back-transformed σ/µ least square means were 0.563 for the airblast and 1.345 for the SSCDS. Comparisons between least square means of the coefficient of variation using Tukey’s adjustment were significantly different from each other in each of the three months, with a higher SSCDS σ/µ at each date: June (t_α = 0.05,36_ = −8.97, adj. *p* < 0.0001), July (t_α = 0.05,36_ = −11.06, adj. *p* < 0.0001), and August (t_α = 0.05,36_ = −12.58, adj. *p* < 0.0001). SSCDS σ/µ displayed significant differences between July and August (t_α = 0.05,36_ = −4.57, adj. *p* = 0.0007) as well as June and August (t_α = 0.05,36_ = −4.68, adj. *p* = 0.0005), while airblast plots did not. (Table 4)

### 3.3. Deposition

Analysis using data from the June, July, and August trials showed significantly higher chemical deposition in SSCDS treated plots than in airblast treated plots (F_1,6_ = 15.84, *p* = 0.0073). The interaction of treatment and height fixed effect terms was also significant (F_2,12_ = 7.41, *p* = 0.0080). Date was also a significant fixed effect (F_2,36_ = 16.41, *p* < 0.0001). Multiple comparisons by month using the Tukey-Kramer adjustment show that SSCDS treated plots displayed significantly higher deposition in all three months: June (t_α = 0.05,36_ = −3.09, adj. *p* = 0.0413), July (t_α = 0.05,36_ = −3.69, adj. *p* = 0.0088), and August (t_α = 0.05,36_ = −3.10, adj. *p* = 0.0402) (Figure 6).

Comparisons between treatments at the same height showed no significant differences between the deposition on the highest sampled leaves (t_α = 0.05,12_ = −1.16, adj. *p* = 0.8450), but leaves at the lower and middle heights showed significantly higher deposition from the SSCDS (t_α = 0.05,12_ = −4.58, adj. *p* = 0.0064; t_α = 0.05,12_ = −4.08, adj. *p* = 0.0149). Different heights in airblast treated plots were not significantly different from each other: high and low (t_α = 0.05,12_ = 2.40, adj. *p* = 0.2296), high and middle (t_α = 0.05,12_ = 1.23, adj. *p* = 0.8132), and middle and low (t_α = 0.05,12_ = −1.21, adj. *p* = 0.8221). Differences in deposition due to height in SSCDS treated plots were also non-significant: high and low (t_α = 0.05,12_ = −2.68, adj. *p* = 0.1519), high and middle (t_α = 0.05,12_ = −281, adj. *p* = 0.1239), and middle and low (t_α = 0.05,12_ = −0.18, adj. *p* = 0.8600).

Coefficients of variation (σ/µ) calculated for each height in each plot for the three dates yielded treatment as a significant effect (F_2,6_ = 28.99, *p* = 0.0017), as well as date (F_2,36_ = 7.84, *p* = 0.0015), but not height (F_2,12_ = 2.16, *p* = 0.1579). There were no significant interactions. Coefficient of variation least square means were 0.7185 ± 0.028 for the airblast and 0.9343 ± 0.028 for the SSCDS. Comparisons between least square means of the coefficient of variation using Tukey’s adjustment were significantly different from each other in June (t_α = 0.05,44_ = −3.79, adj. *p* = 0.0068) and August (t_α = 0.05,44_ = −3.26, adj. *p* = 0.0271), with a higher SSCDS σ/µ, but not in July (t_α = 0.05,44_ = −2.28, adj. *p* = 0.2299). (Table 5)

A separate ANOVA model fitted to just SSCDS deposition data from all four dates (May, June, July, and August) resulted in a significant F-test for the main effect of date (F_3,27_ = 4.91, *p* = 0.0075). Deposition in July and August were both significantly lower than the deposition in June (t_α = 0.05,27_ = −3.28, adj. *p* = 0.0144; t_α = 0.05,27_ = −2.78, adj. *p* = 0.0454), but other date combinations did not display significant differences. Another ANOVA model was fitted to airblast deposition from June, July, and August, and showed the same pattern, with deposition in July and August significantly lower than the deposition in June (t_α = 0.05,18_ = −5.45, adj. *p* < 0.0001; t_α = 0.05,18_ = −3.84, adj. *p* = 0.0033), but July and August were not significantly different from each other (t_α = 0.05,18_ = −1.51, adj. *p* = 0.3087).

### 3.4. Pest Management

Apple scab damage evaluations did not yield any scores higher than 5% on leaves, terminals, or fruit in both treatments (Figure 7). A single sign of leafroller damage was observed in airblast plots. Collected fruit with entries and frass did not yield any live larvae. Wilcoxon rank sum tests between the non-normal airblast and SSCDS plots did not show any significant differences except for the incidence of apple scab on clusters (Figure 8 and Table 6). No apple scab was found on fruit in airblast treated plots, while two incidences of 5% scab damage and 12 incidences of 2% scab damage were found on SSCDS treated fruit. Proportions of damaged fruit and leaves are displayed in Figure 7; Figure 8, along with an untreated comparison.

## 4. Discussion

Coverage evaluations collected in this experiment showed that the prototype SSCDS provides comparable levels of coverage on the adaxial leaf surface to an airblast sprayer (Figure 4). However, SSCDS coverage on the adaxial surface is far higher than the SSCDS coverage on the abaxial surface, which confirms previous observations [8,28,31]. Additionally, coverage on the underside of leaf surfaces is far lower when sprayed with the SSCDS than the coverage obtained with airblast spraying, and significantly lower in almost all cases (Figure 5). Despite the low abaxial coverage, SSCDS sprayed plots exhibited equal or greater levels of chemical deposition on sampled leaves (Figure 6), which implies less chemical was lost from the SSCDS sprays in the form of off target drift. Both systems demonstrated near identical levels of pest control, which is ultimately the most important characteristic of any spray application.

The adaxial coverage measurements only showed an overall significant difference between the two spray types in August, with similar levels of coverage in June and July. Most major sprays in Michigan are applied between mid-April and July, with only three sprays in the test orchard in August (Table 2). This suggests that despite the lower adaxial coverage in August, the SSCDS can provide similar levels of coverage in the major portion of the growing season. However, SSCDS adaxial coverage was significantly lower at the highest sampled height (2.1 m) than the coverage it provided in the lower and middle portions of the canopy, and significantly lower than the adaxial coverage from an airblast at the same height. This sampling height was approximately 2/3 of the height of typical high density apple trees (2.5–3.3 m), and just under the height the highest set of sprinklers sprayed from. Coverage there could potentially be improved by changes in the height, arrangement, or number of the highest set of emitters to attain levels of coverage seen in the lower and middle portion of the canopy. Mean adaxial coverage was still well within the recommended range of coverage found in the literature, from 15% to 30% [23,36], falling within or above this range at all heights and dates.

Coefficients of variation of the adaxial coverage were around two-fold greater in SSCDS sprayed plots, indicating greater heterogeneity in coverage than the airblast. The magnitude of this difference was highest at the top of the canopy, where coverage was the lowest. Coefficients of variation were significantly higher in the SSCDS treatment compared to the airblast treatment at each date, with the biggest differences in August, when the SSCDS also showed significantly lower coverage than the airblast sprayer. This could be attributed to the full canopy development and foliar growth that occurs throughout the summer, blocking spray from the fixed emitters and resulting in lower and uneven coverage.

Abaxial coverage was significantly lower for the SSCDS at every height and on every date overall when compared to the airblast sprayer. It was lowest in the bottom portion of the canopy and the highest portion of the canopy. This is likely because the solid set lacks the moving air front from an axial fan, which both lifts and turns leaves [37]. This action spreads fine droplets within a turbulent airstream so they either intercept the underside of leaves or are sprayed directly onto the upturned abaxial surface. The droplets delivered through the SSCDS are far less likely to reach the underside of the leaf unless it is located near the emitter and received direct spray or natural air movement carries droplets through the canopy. Additionally, SSCDS abaxial coverage exhibited significantly higher coefficients of variation than the airblast (Table 4). In some cases the coefficients of variation was nearly four times greater than the corresponding coefficients of variation seen in airblast treated plots. This was greater than the disparity between coefficients of variation in adaxial coverage as well, showing abaxial surfaces not only receive less coverage, but have far more variable coverage when treated by the SSCDS.

It is important to note that the heterogeneous coverage and deposition referred to here is at the macroscale level of the tree or plot, rather than on the scale of individual droplets. The SSCDS not only exhibits variable coverage at the plot level, but also has a characteristically coarser distribution of droplets intercepting the spray cards. The coarse coverage at the plot level and the droplet level are likely related to some degree, the large splatters or light dusting of droplets on cards lead to much more variable overall percent coverage. However, the plot level heterogeneity of coverage can also be attributed to the static nature of the sprinklers.

Coverage variability on both surfaces and the low coverage on abaxial surfaces is likely caused by inherent properties of fixed spray systems. They may exhibit greater heterogeneity than airblast sprayers since spray interception is much more likely to occur closer to where it is emitted from nozzles. Leaves further from the nozzle are less likely to intercept droplets, especially if they are distant vertically. Larger droplets are subject to gravity rather than air currents, and are pulled downward once they lose momentum [38]. This means adaxial surfaces receive a shower from above, but abaxial surfaces only receive droplets sprayed directly up onto them or the finest droplets that travel through the canopy environment on air currents. Literature has also been published on the local cooling effect provided by microsprinklers, which has been used for sunburn protection in apples [31]. Data collected in this orchard has shown a 2–3 °C drop in temperature immediately following spray applications (Owen-Smith, unpublished). The cool air produced by this effect may also pull spray droplets downward as it sinks, contributing to the lower abaxial coverage and the lower coverage levels seen at the highest sampled portion of the canopy.

Deposition showed a very different profile than coverage: mean SSCDS deposition was greater at every height and date compared to the airblast sprayer. In fact, the lower and middle canopy heights had significantly higher deposition in the SSCDS plot compared with the airblast plots. Coefficients of variation for deposition, while higher in the SSCDS plots compared to the airblast sprayer plots were less pronounced compared to coverage, however, significant differences were detected in June and July. Overall higher levels of deposition in the SSCDS plots suggest that more chemical was retained on the leaf surfaces in SSCDS plots, and less may have been lost to drift.

This hypothesis is consistent with general observations on off-target applications from radial airblast sprayers. Mass balance experiments in dwarf apple trees have shown 10–12% of the spray volume was lost to the ground, and 37–59% lost to the air [13], with 4–17% lost to the air and 10–22% lost to the ground in semi-dwarf trees. A separate study in Italian apple orchards has shown a loss of 37% of the spray to the ground and 7% to the air [39]. Without the moving front of air pushing small droplets above the canopy or into the ground, it is likely that more droplets intercept leaves and are deposited. If 40–50% of the airblast spray was lost to the ground and air, the difference between the mean deposition for each treatment would have been negligible if it had not been off target. This suggest that the SSCDS has the potential to reduce off target deposits, and may serve to reduce drift and soil contamination compared to an orchard airblast sprayer. And in fact, a later study that directly compared vertical and horizontal off target drift for this SSCDS prototype and airblast sprayer which showed a nearly two order of magnitude reduction in off target drift for the SSCDS system compared to the airblast sprayer [40].

Environmental conditions have a great deal of impact on the outcome of spray coverage and deposition. Average wind speed remained under 5 m s^−1^ for each of the spray events, but wind direction was different on each date. However, wind was always directed at an angle across the row, and never completely north/south (Table 1). This may have helped pull spray from row to row in SSCDS treated blocks increasing deposition. Visually, the fixed emitters throw little spray above the row compared to an airblast, and droplets from the airblast may have been caught by these winds and pulled away from the target environment.

Though the SSCDS demonstrated higher deposition, it was distributed less homogenously. Typically, a uniform dispersion of droplets with similar levels of coverage and deposition throughout the canopy is considered ideal and coarse droplet patterns with variable coverage and deposition as something to be avoided as wasteful or inefficient [41,42]. Coarse sprays may result in less optimal coverage than fine sprays [43]. However, Doruchowski et al. also found that air induction nozzles, known for their coarse droplets size and low drift potential, had a similar biological efficacy when compared to fine spray hollow cone nozzles [44]. A concern raised by the heterogeneous distribution of droplets on the micro scale, rather than the macro or plot scale, is phytotoxicity from concentrated agrochemicals. Localized overexposure on leaves that are in close proximity to microsprayers is a potential problem, and would necessitate the selection of compounds where this is a low risk. There was no observed issues with leaf discoloration, malformation, or dropping—but it was not looked for.

Despite what might be considered inferior spray coverage, SSCDS plots exhibited near identical levels of disease and pest management to the airblast sprayer plots, with the only significant difference being a very low incidence of light (<5%) apple scab on fruit. Near identical levels of coverage and insect and disease control were reported over a two year study on the same orchard block for a previous prototype of the SSCDS tested in this study. The tested areas in that study were confined to narrow two row plots, the present study demonstrates that equivalent control can be achieved in larger square plots. In addition, the previous study did not evaluate deposition, relying solely on water sensitive cards to estimate coverage [10]. This highlights the importance of not relying solely on coverage estimates when evaluating spray application technology and suggests that deposition estimates may be a better predictor of pest management efficacy. Another recent study evaluating coverage and deposition provided by a prototype SSCDS in vineyards showed a similar pattern of lower coverage but higher deposition, but did not evaluate pest management [45]. There are several possible explanations for the consistent disparities between coverage and deposition provided by SSCDS. One potential explanation is that fine droplets below the detectable threshold of either the WSP or scanner contributed to coverage. Verpont et al. hypothesized these tiny droplets may still provide enough chemical residue for biological activity [28]. A second explanation is that the higher deposition provided by the SSCDS was redistributed during rewetting periods (e.g., dew and rainfall events) leading to enhanced coverage over time. Prior studies have also reported there is often minimal correlation between the observed deposition profile and the actual biological efficacy of the spray [44], and that coverage and pest control do not necessarily correspond with each other [46]. Pest management is the true goal of any application, and these results support the solid set canopy delivery system’s potential as an alternative to airblast sprayers.

Despite its proven ability for pest management, there are still some concerns raised by heterogeneous coverage. Potential issues may arise with pests or pathogens that reside on the underside of the leaf-where the SSCDS has inferior coverage. Reservoirs of fungal bodies or spores may also avoid treatment if they are sheltered from treatment by dense foliage occluding spray. For example, Viret et al. showed powdery mildew control in vines was best when both sides of the leaf received near equal treatment [46]. Research has demonstrated that suitable coverage patterns are partly dependent on the mode of action of the compound [24,47]. Many of the modern pesticides used in apple IPM programs hold plant penetrative properties that allow translaminar movement of compounds from adaxial to abaxial leaf surfaces [48]. For these materials, coverage is less important than deposition to provide the expected plant protection. Arthropods such as European Red Mite (*Panonychus ulmi*, Koch) require near complete coverage for control since they lay eggs in crevices and spend much of their time on the underside of the leaf [49]. The SSCDS could conceivably have issues controlling pests that have concealed life stages or aren’t motile, or that manage to avoid areas that receive spray. On the other hand, pests such as apple maggot (*Rhagoletis pomonella*, Walsh) adults are very active and don’t require high levels of coverage to receive lethal exposure to toxicant. Thus, the efficacy of SSCDS is likely determined by coverage and deposition, the chemistry used, pest targeted, and environmental conditions.

Further work is needed to evaluate the management potential of SSCDS for other growing environments, such as in the near desert conditions of Washington State USA. Studies on the drift profile of SSCDS systems are also needed to evaluate whether these systems do in fact reduce off target loss of product. Probably the most pressing need for developing this technology is the development of specialized microsprayers optimized for agrochemical delivery and their arrangements in the canopy. Singha et al. (2019) showed that the incorporation of hollow cone nozzles as well as positioning microsprinklers so that they spray upward from the base of the canopy improved both the amount and uniformity of coverage, particulary abaxial coverage, in vineyards [45]. Finally, further research in different perennial crops (e.g., blueberries, cherries, stone fruit, etc.) are needed so that we can better understand canopy architectures that are compatible with this exciting, potentially disruptive agrochemical delivery technology.

## 5. Conclusions

A prototype SSCDS provided comparable season-long pest management to an airblast sprayer although coverage and deposition varied greatly between the two agrochemical delivery platforms. Both coverage and deposition were more variable in the SSCDS suggesting that it is a less consistent application technology. Adaxial percentage coverage was largely comparable between the two systems but abaxial coverage was much lower in the SSCDS. Deposition, as estimated with tartrazine tracer dye was higher in the SSCDS. The increased canopy deposition in the SSCDS system suggests that this system may produce less off-target drift compared with a radial fan airblast sprayer. These results support the further development of SSCDS systems for high density apples.

## Figures and Tables

**Figure 1 insects-10-00193-f001:**
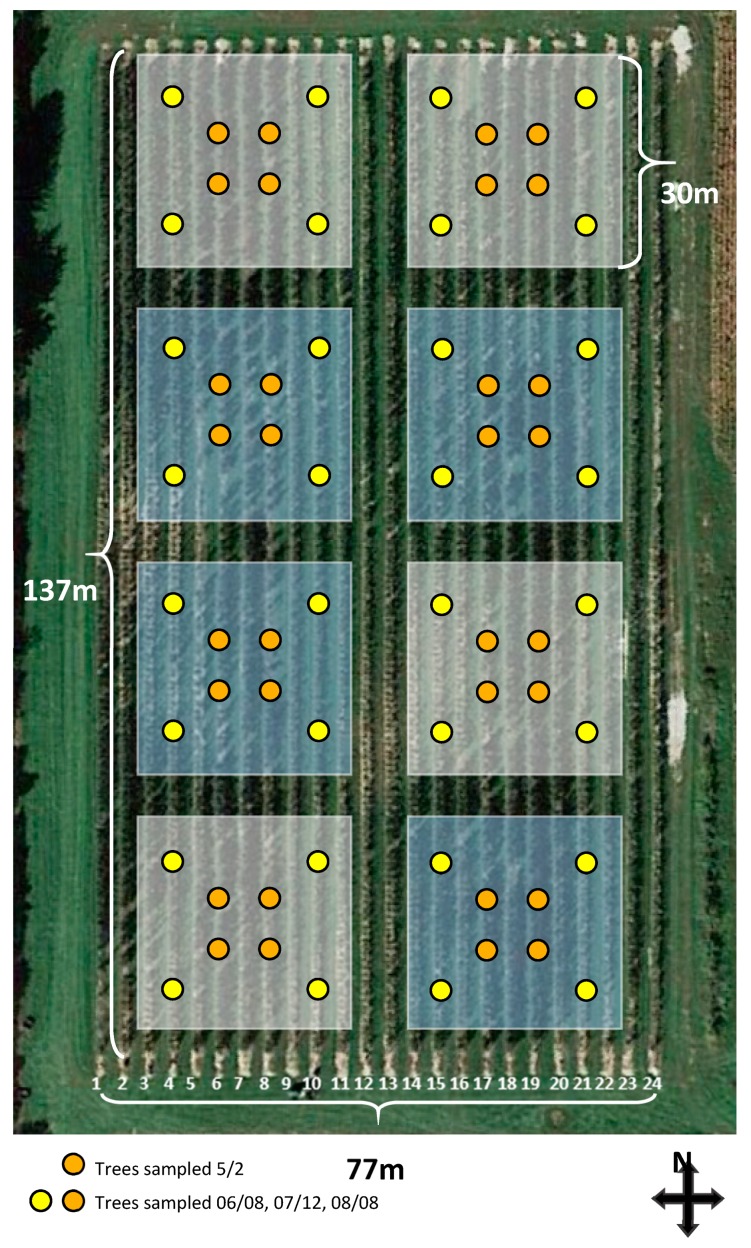
Plot plan of the SSCDS orchard installation. Blue shaded squares denote SSCDS sprayed plots, white squares are airblast plots. Yellow and orange markers are the approximate location of sampled trees.

**Figure 2 insects-10-00193-f002:**
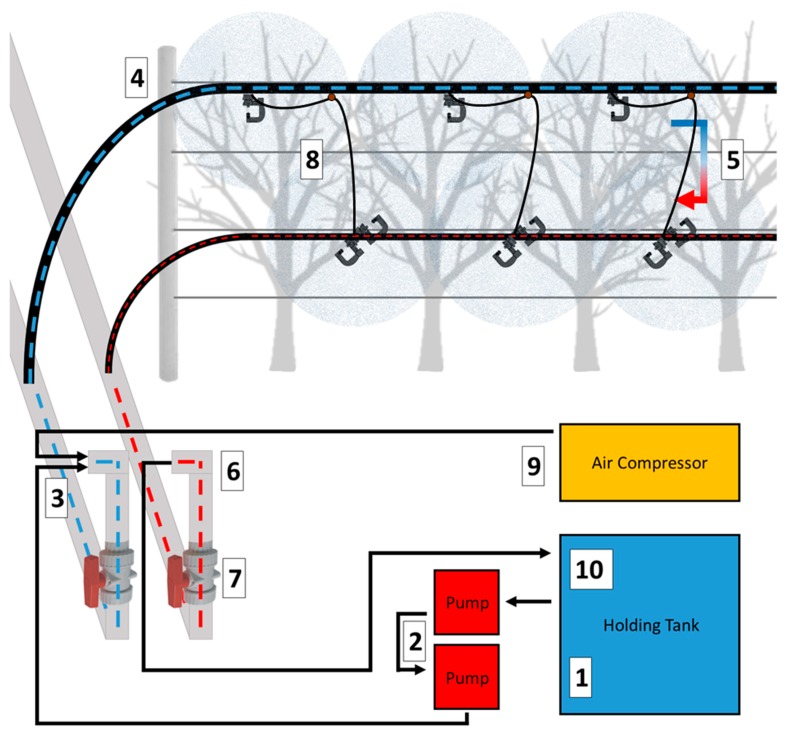
Schematic of Prototype Solid Set Canopy Delivery System.

**Figure 3 insects-10-00193-f003:**
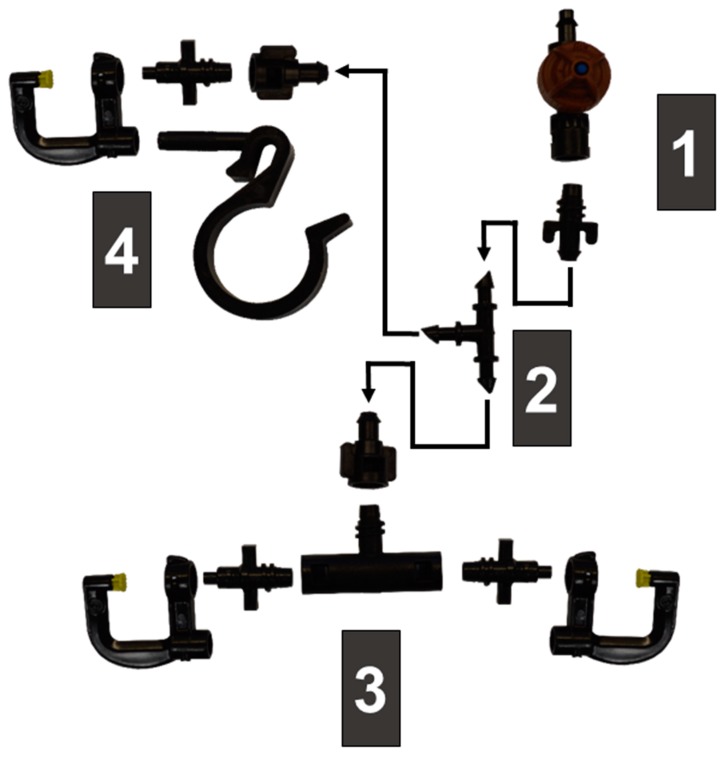
Microsprinkler components for a set of sprayers. (1) 241 kPa Leak prevention device that is inserted into 2.5 cm upper line with bayonet adapter to 0.635 cm line. (2) 3 way split inserted into 0.635 cm line. (3) ‘T’ bayonet fitting attached to a pair of 0.8 mm nozzles and Hadar 7110 spray deflectors. (4) Single microsprinkler fixed to post on hose/wire clip.

**Figure 4 insects-10-00193-f004:**
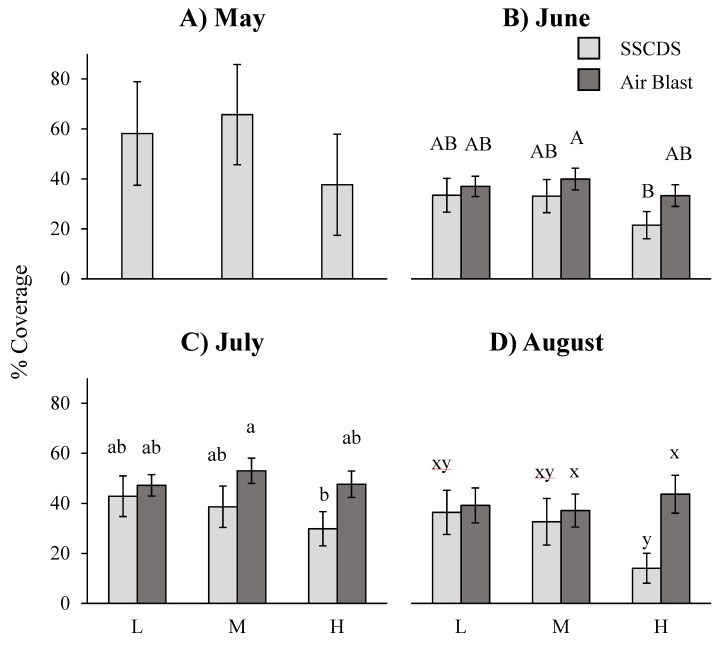
Least square means of the percent adaxial coverage (cm^2^). Means shown for each of the sampled dates (**A**) 5/2/16, (**B**) 6/8/16, (**C**) 7/12/16, and (**D**) 8/8/16) and heights (L = 0.7 m, M = 1.4 m, H = 2.1 m). Error bars indicate (Tukey-Kramer) 95% confidence intervals. Different letters within a month indicate significant differences among bars within a month. No significant differences were detected in May (SSCDS system only).

**Figure 5 insects-10-00193-f005:**
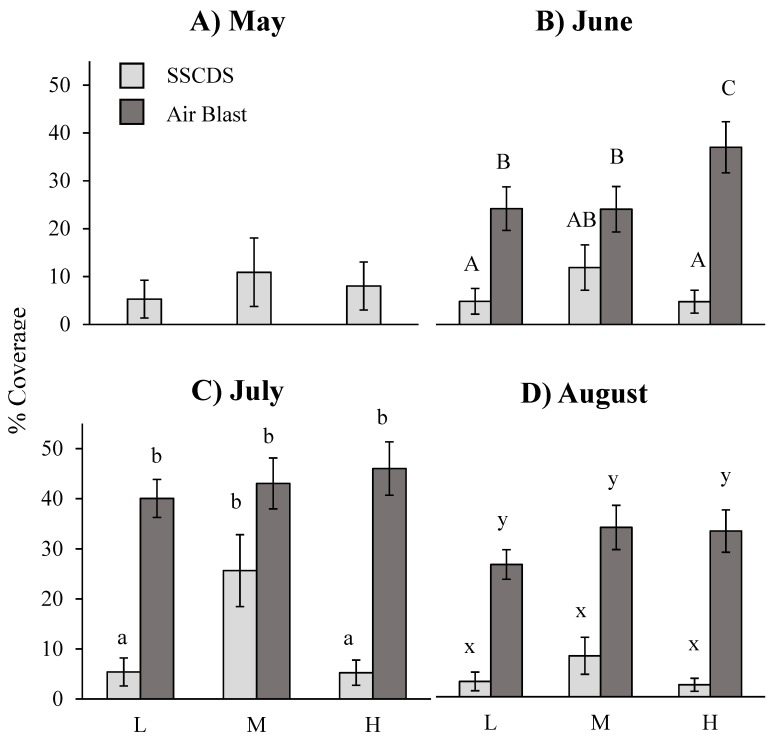
Least square means of the percent abaxial coverage (cm^2^). Means shown for each of the sampled dates (**A**) 5/2/16, (**B**) 6/8/16, (**C**) 7/12/16, and (**D**) 8/8/16 and heights (L = 0.7 m, M = 1.4 m, H = 2.1 m). Error bars indicate (Tukey-Kramer) 95% confidence intervals. Different letters within a month indicate significant differences among bars within a month. No significant differences were detected in May (SSCDS system only).

**Figure 6 insects-10-00193-f006:**
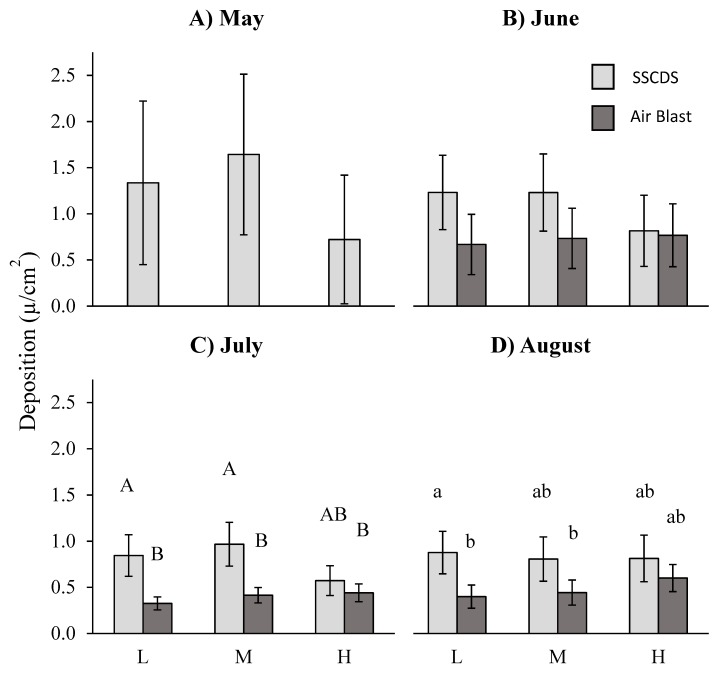
Least square means of deposition of tartrazine dye (µg/cm^2^). Means shown for each of the sampled dates (**A**) 5/2/16, (**B**) 6/8/16, (**C**) 7/12/16, and (**D**) 8/8/16 and heights (L = 0.7 m, M = 1.4 m, H = 2.1 m). Error bars indicate (Tukey-Kramer) 95% confidence intervals. Different letters within a month indicate significant differences among bars within a month. No significant differences were detected in May and June.

**Figure 7 insects-10-00193-f007:**
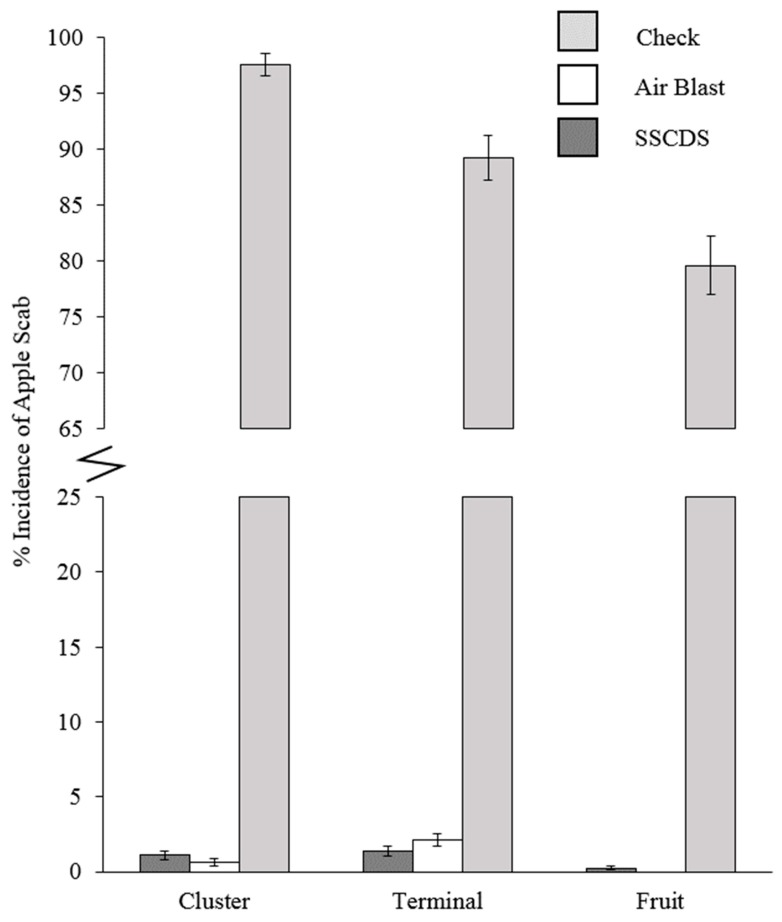
Observed percent incidence of apple scab in each tissue type. Mean incidence shown from the airblast and SSCDS treated plots, with a comparison check from a nearby untreated orchard (Check not included in statistical analysis). Error bars indicate standard error of the proportion.

**Figure 8 insects-10-00193-f008:**
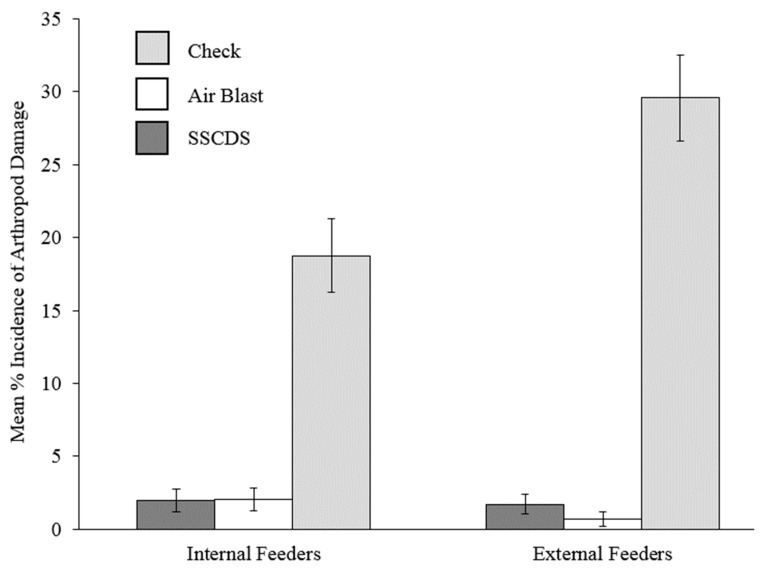
Mean percent incidence of arthropod damage on fruit. Mean incidence shown from the airblast and SSCDS treated plots, with a comparison check from a nearby untreated orchard (Check not included in statistical analysis). Error bars indicate standard error of the proportion.

**Table 1 insects-10-00193-t001:** Environmental weather data from Clarksville Research Station’s Enviro-weather station on spray evaluation days. Wind came from the E/NE in May, W/NW in June, S/SW in July, and E/SE in August.

Date	Time	Air Temp (°C)	Precip. (mm)	Relative Humidity (%)	Wind Speed (m/s)	Max Wind Speed (m/s)	Wind Direction (° from N)
	10:00	6.8	0	83.3	0.6	2.9	54.3
	11:00	8.0	0	77.2	0.9	2.5	67.7
	12:00	9.6	0	74.7	0.8	2.8	60.0
2-May-2016	13:00	10.3	0	72.7	0.7	2.6	81.8
	14:00	11.7	0	69.1	0.8	2.9	60.9
	15:00	12.7	0	66.2	0.6	2.5	68.7
	16:00	12.6	0	66.8	0.5	2.3	76.4
	10:00	11.4	0	70.9	3.3	6.5	299.7
	11:00	12.8	0	59.1	3.8	7	296.2
	12:00	14.1	0	53.4	3.3	6.5	287.2
6-June-2016	13:00	15.1	0	51	3.3	6.7	294.5
	14:00	16.2	0	49.8	3.4	7.4	291.1
	15:00	17.3	0	47.4	3.2	6.5	297.0
	16:00	18.2	0	44.1	4.1	9.1	287.0
	10:00	24.8	0	70.1	2.3	5.8	197.7
	11:00	26.6	0	65.6	2.4	5.2	198.7
	12:00	27.9	0	63.9	2.7	6.2	200.4
12-July-2016	13:00	29.0	0	61.3	2.4	5.8	189.5
	14:00	30.4	0	55.8	2.5	7.1	184.9
	15:00	30.1	0	54.8	2.4	6.7	182.5
	16:00	31.1	0	51.8	2.8	8.4	189
	10:00	19.6	0	81.9	0.5	2.1	106.8
	11:00	22.6	0	64	0.7	2.6	131.3
	12:00	24.8	0	52.1	0.8	2.8	95.5
8-August-2016	13:00	26.1	0	46.2	0.7	3	84.0
	14:00	26.6	0	42	0.7	2.8	104.1
	15:00	27.5	0	39.4	0.8	3.4	103.1
	16:00	27.9	0	39.2	0.7	2.9	62.6

**Table 2 insects-10-00193-t002:** Schedule of pesticide applications made through the airblast sprayer and SSCDS system in for season long pest management.

Date	Product	Type	Active Ingredient	Rate
29-Apr-2016	Sivanto Prime L	Insecticide	Flupyradifurone	0.88 L/ha
	Manzate Pro-Stik	Fungicide	Zinc ion and manganese ethylenebisdithiocarbamate	6.73 kg/ha
	Aprovia	Fungicide	Benzovindiflupyr	0.31 L/ha
6-May-2016	Manzate Pro-Stik	Insecticide	Zinc ion and manganese ethylenebisdithiocarbamate	4.48 kg/ha
	Inspire Super	Fungicide	Difenoconazole	0.62 L/ha
13-May-2016	Aprovia	Fungicide	Benzovindiflupyr	0.31 L/ha
	Roper	Fungicide	Zinc ion and manganese ethylenebisdithiocarbamate	4.48 kg/ha
	Kasumin	Bactericide	Kasugamycin Hydrochloride Hydrate	4.68 L/ha
19-May-2016	Assail	Insecticide	Acetamiprid	0.44 L/ha
	Rally	Fungicide	Myclobutanil	0.37 L/ha
25-May-2016	Luna Sensation	Fungicide	Fluoopyram and trifloxystrobin	0.37 L/ha
	Belay	Insecticide	Clothianidin	0.29 L/ha
6-Jun-2016	Ziram	Fungicide	Zinc dimethyldithiocarbamate	5.6 kg/ha
	Rally	Fungicide	Myclobutanil	0.37 L/ha
	Reaper	Insecticide	Abamectin	0.73 L/ha
	Prey	Insecticide	Imidacloprid	0.44 L/ha
	Belay	Insecticide	Clothianidin	0.44 L/ha
	Belt	Insecticide	Flubendiamide	0.29 L/ha
	Damoil	Insecticide	Mineral Oil	9.35 L/ha
14-Jun-2016	Ziram	Fungicide	Zinc dimethyldithiocarbamate	4.48 kg/ha
	Assail	Insecticide	Acetamiprid	0.47 L/ha
29-Jun-2016	Flint	Fungicide	Trifloxystrobin	0.15 L/ha
	Captan Gold	Fungicide	N-Trichloromethylthio-4-cyclohexene-1,2-dicarboximide	5.6 kg/ha
	Altacor	Insecticide	Chlorantraniliprole	0.29 L/ha
8-Jul-2016	Movento	Insecticide	Spirotetramat	0.66 L/ha
19-Jul-2016	Nealta	Miticide	Cyflumetofen	1 L/ha
	Captan Gold	Fungicide	N-Trichloromethylthio-4-cyclohexene-1,2-dicarboximide	3.36 kg/ha
22-Jul-2016	Indar	Fungicide	Fenbuconazole	0.44 L/ha
1-Aug-2016	Delegate	Insecticide	Spinetoram	0.32 L/ha
	Flint	Fungicide	Trifloxystrobin	0.15 L/ha
15-Aug-2016	Delegate	Insecticide	Spinetoram	0.32 L/ha

**Table 3 insects-10-00193-t003:** Coefficient of Variation of adaxial coverage (cm^2^) for each date and height. Least square mean of each treatment, height, and date combination on the left and least square mean of the overall date and treatment combination to the right. Asterisks denote a significant difference between months within a treatment.

		June		July		August	
**Airblast**	L	0.493		0.381		0.560	
	M	0.502	0.529	0.419	0.434 *	0.536	0.5744 *
	H	0.593		0.503		0.627	
**SSCDS**	L	0.830		0.816		0.883	
	M	0.876	0.9397	0.902	0.908	1.014	1.261
	H	1.112		0.984		1.481	

**Table 4 insects-10-00193-t004:** Coefficient of Variation of abaxial coverage (cm^2^) for each date and height. Least square mean of each treatment, height, and date combination on the left and least square mean of the overall date and treatment combination to the right.

		June		July		August	
**Airblast**	L	0.598		0.346		0.566	
	M	0.712	0.619	0.521	0.458	0.655	0.6201
	H	0.551		0.517		0.642	
**SSCDS**	L	1.924		2.082		2.448	
	M	1.394	1.631	1.142	1.644	1.915	2.179
	H	1.587		1.782		2.192	

**Table 5 insects-10-00193-t005:** Coefficient of Variation of deposition (µg/cm^2^) for each date and height. Least square mean of each treatment, height, and date combination on the left and least square mean of the overall date and treatment combination to the right.

		June		July		August	
**Airblast**	L	0.186		0.186		0.300	
	M	0.204	0.586	0.236	0.777	0.282	0.792
	H	0.327		0.290		0.303	
**SSCDS**	L	0.639		0.701		0.704	
	M	0.893	0.8494	0.660	0.935	0.833	1.02
	H	0.729		0.629		0.976	

**Table 6 insects-10-00193-t006:** Results of four exact Wilcoxon rank sum tests for each category of damage. ‘Internal’ refers to internally feeding lepidopteran damage from stings or entries, and ‘external’ refers to damage from Plum Curculio and Pentatomidae. ‘Terminal’ and ‘cluster’ refer to apple scab damage on those portions of the plant. Significance was determined at the alpha = 0.05 level.

Damage Type		Z-Value	Pr. < Z
Arthropod	Internal	−1.080	0.140
	External	1.527	0.063
Apple Scab	Terminal	0.540	0.295
	Cluster	−0.588	0.278

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
