# Peer review of "Season Long Pest Management Efficacy and Spray Characteristics of a Solid Set Canopy Delivery System in High Density Apples"

_insects, 2019, doi:10.3390/insects10070193_

Reviewer 1 Report

This is an excellent and very useful paper. The suggestions I have here are not hard and fast, required edits- this paper could be published as-is without sacrificing the inegrity of thre joural or the authors.

Inline comments:

Plot layout appears appropriate, measures were taken to minimize drift and collection of readings within drift-potential areas.

SSCDS design looks great. What is the purpose of ganging two pumps together? Increased pressure?

Lines 207-219: Has this method been used in other studies? It appears sound, but comparison to others might be helpful.

232: Is that rate of 5.3 kg/ha tebuconazole right? It seems awfully high.

Table 2: Five insecticides applied on June 6? That seems like overkill, but it’s not really part of the methods to question what was sprayed. However, some reference to the IPM decision-making process would be helpful.

240-260: Can these methods be summarized and supported by another citable publication? Can you reference observer training or other measures taken to ensure consistency in ‘calling’ damage?

256-260: This was a good exercise to go through, but isn’t statistically valid.

2.3.4 Stats methods look sound. Appropriate assumption tests were conducted. Somewhere you should mention the α level that you use, I assume 0.05 but that’s not always what is used.

Figures 4-6: The mean separation letters are a little hard to figure out, what As go with what ABs and Bs. I was able to sort it out through a careful reading of the results and the figure captions, but it was not intuitive. No hard recommendation made for correction, but if there is another way to phrase or visualize, it may be helpful.

Table 3 should indicate the differences mentioned in the text between July & August for SSCDS. Should also reference what the original units used for adaxial coverage were.

409-416: Follow-up ANOVA models for deposition within each treatment are appreciated and strengthen the paper.

Pest management: did you use Wilcoxon rank sum test because the mode zero data failed homogeneity/normality assumptions? Say so if so.

Figure 8: Mean percent of arthropod damage on fruit

450-451: That’s the critical point to make. May want to ponder issues with heterogeneity of distribution, e.g. if all the material is on top of the leaves, could we get some phytotoxicity?

503-506: Good observation of the cooling effect and hypothesis re: it’s effect on droplet and air movement.

554-556: I have been thinking about redistribution of material during wetting events and how it could ameliorate heterogeneous distribution in SSCDS through much of this paper. Good job highlighting it in thew discussion.

Author Response

SSCDS design looks great. What is the purpose of ganging two pumps together? Increased pressure?

Clarified on lines 131-132.

Lines 207-219: Has this method been used in other studies? It appears sound, but comparison to others might be helpful.

Our method was developed based on advice by Mark Ledebuhr a private spray technologist. The particulars are not published.

232: Is that rate of 5.3 kg/ha tebuconazole right? It seems awfully high.

This was a conversion error (English to metric) on our part. Thanks for catching it!!

Table 2: Five insecticides applied on June 6? That seems like overkill, but it’s not really part of the methods to question what was sprayed. However, some reference to the IPM decision-making process would be helpful.

We used a spray program typical of MI apple production –the program used is the management program utilized by the research farm. The June spray window targets multiple arthropods and diseases (internal and external feeding moths, plum curculio, aphids, mites and apple scab) and with the loss of Guthion (Azinphosmethyl) growers are increasingly using mixes of insecticides (compounds have become more taxa specific).

240-260: Can these methods be summarized and supported by another citable publication? Can you reference observer training or other measures taken to ensure consistency in ‘calling’ damage?

We’ve added a citation for the extension publication used to train our observers. LN 273

256-260: This was a good exercise to go through, but isn’t statistically valid.

The purpose of the check was to show that unmanaged apples in this landscape are heavily damaged by the pests evaluated in our replicated plots –we have tried to reinforce this in the text and figure captions. LN 281-284.

2.3.4 Stats methods look sound. Appropriate assumption tests were conducted. Somewhere you should mention the α level that you use, I assume 0.05 but that’s not always what is used.

Good catch! We’ve added our alpha value LN: 301-302

Figures 4-6: The mean separation letters are a little hard to figure out, what As go with what ABs and Bs. I was able to sort it out through a careful reading of the results and the figure captions, but it was not intuitive. No hard recommendation made for correction, but if there is another way to phrase or visualize, it may be helpful.

We’ve added subfigure letters to try and make this easier to interpret and have reworded the figure caption to reflect the patterns of significance across them.

Table 3 should indicate the differences mentioned in the text between July & August for SSCDS. Should also reference what the original units used for adaxial coverage were.

We have made the requested change.

409-416: Follow-up ANOVA models for deposition within each treatment are appreciated and strengthen the paper.

Thank you! We made a real effort to use appropriate analyses.

Pest management: did you use Wilcoxon rank sum test because the mode zero data failed homogeneity/normality assumptions? Say so if so.

This was specified in the statistical methods section LN 313-316. We have added text to the results to further clarify this. LN: 450

Figure 8: Mean percent of arthropod damage on fruit

We have applied this change

450-451: That’s the critical point to make. May want to ponder issues with heterogeneity of distribution, e.g. if all the material is on top of the leaves, could we get some phytotoxicity?

We were not sure how to address this comment. It is a good question but we did not see any evidence of phytotoxicity in our trial so we don’t feel it is appropriate to discuss it.

503-506: Good observation of the cooling effect and hypothesis re: it’s effect on droplet and air movement.

Thank you! We spent a lot of time thinking about the seeming disconnect among our coverage, deposition and pest management outcomes.

554-556: I have been thinking about redistribution of material during wetting events and how it could ameliorate heterogeneous distribution in SSCDS through much of this paper. Good job highlighting it in the discussion.

Thank you! We agree and will be performing some pilot studies this year to evaluate just this phenomenon.

Reviewer 2 Report

Overall, the paper is very well written and depicts and important study that hopefully foreshadows the future of spray technology. Other than a few stylistic and formatting issues, my concerns are 1. check plot(s?) are seemingly invalid, unorthodox and/or not described thoroughly, 2. methods for insect and pathogen sampling need more detail, and 3. the study was only conducted in a single season (okay for engineering, not for field entomology). 

Because the check is in a different orchard location and variety, it is important to reassure the reader that the check, while not ideal, was both necessary and valid. 

This journal is called "Insects"; so I think it is important that this study take more time to explain how insect sampling was performed.

While the primary focus is to exhibit spray technology, insect field data really should be collected over at least two seasons to meet the standard for peer review publication. If the authors believe it is unnecessary for this particular study, they should explain why in the paper.

(see attached document with further comments)

Author Response

Overall, the paper is very well written and depicts and important study that hopefully foreshadows the future of spray technology. Other than a few stylistic and formatting issues, my concerns are 1. check plot(s?) are seemingly invalid, unorthodox and/or not described thoroughly, 2. methods for insect and pathogen sampling need more detail, and 3. the study was only conducted in a single season (okay for engineering, not for field entomology). 

1.     In the present study we elected to use check plots because we felt testing the hypothesis that a lack of pest management results in heavy pest damage was unneeded but did want to provide evidence that untreated apple plots in our landscape are heavily damaged by the pests evaluated in our study. We have tried to make it clear that our check plots were not used in quantitative analysis but are provided as a qualitative check. LN: 281-284 and figure captions

2.     We respectfully disagree. We have provided thorough descriptions of not only what data were collected and how they were collected, but also detailed descriptions of the spatial dispersion of sampling within the plot. We have added the online reference used to train our observers. LN: 273.

3.     This study’s pest management outcomes are further supported by our recently published (and cited) Journal of Pest Management Science paper (# 10) which provides 2 consecutive years of nearly identical pest management data including true replicated untreated controls. The major difference of the two studies is in the former study treatments were applied to narrow rows while this study applied treatments to close to square extents. We have tried to further communicate this in LN: 577-582

Because the check is in a different orchard location and variety, it is important to reassure the reader that the check, while not ideal, was both necessary and valid. 

The purpose of the check was to demonstrate that unmanaged apples in this landscape are heavily damaged by the pests evaluated in our replicated plots –we have tried to reinforce this in the text. LN 281-284. In designing our experiment, we decided to use our available resources to develop a strong comparison of pest management potential of SSCDS vs. airblast sprayer programs, rather than include a true untreated control. Inclusion of such a control would have limited our plot size and largely only tested the hypothesis that “unmanaged apples experience heavy pest damage.” It is our strong opinion that that hypothesis has been well established in our growing region as well as those across the globe.

This journal is called "Insects"; so I think it is important that this study take more time to explain how insect sampling was performed.

The reviewer has correctly observed that this research exists at the nexus of spray technology engineering and pest management science. While it is true that the journal is named Insects, the special issue we submitted to was “Pest Control in Fruit Trees” The focus of this paper is the comprehensive end-of-season tree fruit pest management results of an experimental pest management technology with special consideration of how coverage and deposition may impact insect and disease management in apples. We do not agree that the addition of data on the population dynamics, behavior or incidence of individual insect pests add to the story presented in our study and feel that our paper is extremely well suited to the special topics issue it was submitted/developed for.

While the primary focus is to exhibit spray technology, insect field data really should be collected over at least two seasons to meet the standard for peer review publication. If the authors believe it is unnecessary for this particular study, they should explain why in the paper. 

We have added text in the discussion further comparing our results with our previously published work that included both replicated untreated controls and two years of consistent data. LN: 575-581